# Posttranslational Acylations of the Rat Brain Transketolase Discriminate the Enzyme Responses to Inhibitors of ThDP-Dependent Enzymes or Thiamine Transport

**DOI:** 10.3390/ijms25020917

**Published:** 2024-01-11

**Authors:** Vasily A. Aleshin, Thilo Kaehne, Maria V. Maslova, Anastasia V. Graf, Victoria I. Bunik

**Affiliations:** 1Belozersky Institute of Physicochemical Biology, Lomonosov Moscow State University, 119234 Moscow, Russia; aleshin_vasily@mail.ru (V.A.A.); nastjushka@gmail.com (A.V.G.); 2Department of Biochemistry, Sechenov University, 119048 Moscow, Russia; 3Institute of Experimental Internal Medicine, Otto von Guericke University, 39106 Magdeburg, Germany; kaehne@med.ovgu.de; 4Faculty of Biology, Lomonosov Moscow State University, 119234 Moscow, Russia; maslova_masha@mail.ru; 5Faculty of Bioengineering and Bioinformatics, Lomonosov Moscow State University, 119234 Moscow, Russia

**Keywords:** transketolase, malonylation, acetylation, sirtuin 5, oxythiamine, metformin, amprolium, thiamine, posttranslational modifications

## Abstract

Transketolase (TKT) is an essential thiamine diphosphate (ThDP)-dependent enzyme of the non-oxidative branch of the pentose phosphate pathway, with the glucose-6P flux through the pathway regulated in various medically important conditions. Here, we characterize the brain TKT regulation by acylation in rats with perturbed thiamine-dependent metabolism, known to occur in neurodegenerative diseases. The perturbations are modeled by the administration of oxythiamine inhibiting ThDP-dependent enzymes in vivo or by reduced thiamine availability in the presence of metformin and amprolium, inhibiting intracellular thiamine transporters. Compared to control rats, chronic administration of oxythiamine does not significantly change the modification level of the two detected TKT acetylation sites (K6 and K102) but doubles malonylation of TKT K499, concomitantly decreasing 1.7-fold the level of demalonylase sirtuin 5. The inhibitors of thiamine transporters do not change average levels of TKT acylation or sirtuin 5. TKT structures indicate that the acylated residues are distant from the active sites. The acylations-perturbed electrostatic interactions may be involved in conformational shifts and/or the formation of TKT complexes with other proteins or nucleic acids. Acetylation of K102 may affect the active site entrance/exit and subunit interactions. Correlation analysis reveals that the action of oxythiamine is characterized by significant negative correlations of K499 malonylation or K6 acetylation with TKT activity, not observed upon the action of the inhibitors of thiamine transport. However, the transport inhibitors induce significant negative correlations between the TKT activity and K102 acetylation or TKT expression, absent in the oxythiamine group. Thus, perturbations in the ThDP-dependent catalysis or thiamine transport manifest in the insult-specific patterns of the brain TKT malonylation and acetylations.

## 1. Introduction

TKT is a thiamine diphosphate (ThDP)-dependent enzyme of the non-oxidative branch of the pentose phosphate pathway (PPP), where ribose-5-phosphate for cellular nucleotide biosynthesis is generated in the oxidative and non-oxidative branches [1,2,3,4]. TKT is a homodimer, with both subunits (each about 70 kDa) contributing the amino acid residues for each of the two active sites [5]. TKT regulation by posttranslational modifications is suggested by the long-known existence of multiple forms of TKT, which are not due to genetic or splicing factors but differ in the enzyme charge and affinity to ThDP [6,7,8]. The electrostatics-affecting modifications of metabolic enzymes by acylations of their lysine residues, such as the long-known acetylation of histones, are suggested by the recent discovery of an interplay between TKT and an NAD^+^-dependent deacylase of negatively charged acyls, sirtuin 5 [1,9]. For example, in adipose-derived mesenchymal stem cells, the sirtuin 5 knockout significantly up-regulates TKT expression [9]. In colorectal cancer cells, blocking the sirtuin 5 expression does not affect the expression of TKT but decreases the TKT activity; the decrease is accompanied by impaired availability of ribose-5P for nucleotide synthesis and alleviated by overexpression of TKT in the sirtuin-5-silenced cells [1]. Contrary, overexpression of sirtuin 5, but not its catalytically inactive mutant, increases the cellular TKT activity in the colorectal cancer cells [1]. Overall, the data on the cancer cells indicate that malonylation of TKT inactivates the enzyme, which causes DNA damage due to insufficiency of ribose-5P [1]. Yet in the liver of the sirtuin 5 knockout mice, the inhibition of glycolysis is primarily caused by increased malonylation of glyceraldehyde 3-phosphate dehydrogenase, although malonylation of TKT is increased, too [2]. The accumulating data suggest that modification of TKT by the negatively charged acyls removed by sirtuin 5 may result in the known TKT isoforms affecting the enzyme charge and function [8]. The varied responses to the knockout observed in different cells manifest dependence of the responses on specific homeostatic networks. However, the cell-specific regulation and metabolic impact of the posttranslational modifications of TKT have been characterized only in the artificially created systems with sirtuin 5 and TKT knockouts or knockdowns [1,2,9]. Therefore, the question on the naturally occurring levels of the TKT acylations and their regulatory potential in the absence of genetic manipulations arises. In this regard, the goal of our study is to characterize the naturally occurring TKT modifications and their role in cellular homeostatic responses, which are induced by metabolic challenges instead of genetic manipulations. We focus our study on TKT in the brain cortex tissue, in view of the specific significance of glucose metabolism in the brain and the role of TKT and its isoenzyme in neurodevelopment [10,11]. As metabolic challenges, we use the two models of a medically important condition, such as deficient functions of the ThDP-dependent enzymes, central for glucose metabolism. Oxythiamine (OT) is a classic inhibitor of these functions in vivo; as in living systems, it is converted into oxy-ThDP, blocking the ThDP-dependent enzymatic active sites with high affinity [12]. It is also a naturally occurring “damaged” metabolite arising upon the thiamine oxidation under oxidative stress conditions in vivo, particularly associated with chronic kidney disease or kidney transplantation [13,14,15]. Another known thiamine antagonist, amprolium, is a coccidiostatic [16,17], which blocks intracellular transport of thiamine through ThTR-1 and OCT transporters [12,15]. Inhibition of the thiamine transport is also known for the antidiabetic drug metformin, blocking cellular thiamine transporters OCT1 and OCT2 better than amprolium, in addition, inhibiting ThTR-2 [15,18]. As a result, we use these different types of altered thiamine metabolism in rats to assess the functional significance of posttranslational modifications of the rat brain TKT in response to (i) the inhibition of intracellular thiamine transport by a combination of metformin and amprolium (M + A) and (ii) the inhibition of the ThDP-dependent enzymes upon administration of OT. Mass-spectrometry (MS) quantifications of the TKT-acylated peptides in the rat brain homogenates show both specific and common features of the alterations in the brain TKT acylations by different inhibitors of thiamine-dependent metabolism.

## 2. Results

### 2.1. Effect of Inhibitors of Thiamine-Dependent Metabolism on TKT Acylations in the Rat Brain

Three acylated peptides of TKT detected by MS in the rat brain homogenates reveal malonylation of K499 residue and acetylations of K102 and K6 residues in the brain TKT (Appendix A). To quantify relative abundances of these peptides upon the metabolic challenges, their levels are normalized to the same (K102- and K499-containing peptides) or independent (K6-containing peptide) non-acylated TKT peptides, as described in “Materials and Methods”, employing an approach developed and successfully used earlier [19,20,21,22].

Chronic administration of the in vivo inhibitor of the ThDP-dependent enzymes OT for 30 days causes a 2-fold increase (*p* < 0.01) in the malonylation of TKT K499 residue in the rat cerebral cortex (Figure 1). Concomitantly, the level of sirtuin 5 is decreased 1.7-fold (*p* = 0.01) by OT (Figure 1). Since sirtuin 5 is a deacylase-specific for negatively charged acyl groups, such as the malonyl group, the observed decrease in sirtuin 5 level agrees well with the simultaneous accumulation of TKT malonylation (Figure 1). No statistically significant effects of OT are observed on the levels of the TKT acetylations at either K6 or K102 residues (Figure 1).

Chronic administration of the thiamine transporters inhibitors M + A for 30 days has no statistically significant effects on the brain levels of TKT acylations or sirtuin 5 expression (Figure 1).

### 2.2. Positions of the Brain TKT Acylation Sites in the Resolved Structures of Mammalian TKT

In order to assess the potential functional significance of the detected posttranslational modifications of TKT, positions of the acylated residues in the TKT dimer are determined using the structures of mammalian TKT with PDB identifiers 3OOY and 4KXX [23]. The structure 3OOY shows the TKT dimer with bound ThDP. In the structure 4KXX, the TKT N-terminal peptide and the substrate analog hexahydroxyheptyl dihydrogen phosphate are bound in the active site.

As shown in Figure 2, all the three residues undergoing acylations in the brain, namely, K6, K102, and K499, are neither located in the active site nor do they directly participate in ThDP binding. Rather, the residues belong to the surface of the TKT dimer, although K102 is less exposed to the solvent than K6 or K499.

The TKT N-terminal K6 residue undergoing acetylation is localized within the active-site-comprising N-terminal domain of TKT (Figure 2A,B). In the resolved structure 4KXX, the deacetylated positively charged K6 residue forms a salt bridge with the E222 residue, as shown in Figure 2. In another structure of TKT (3OOY), where the N-terminus is not resolved, E222 interacts with K16, localized in the first N-terminal alpha-helix of TKT (Figure 2B). Since lysine acetylation removes the positive charge of the lysine residues, the acetylation of K6 should disable its electrostatic interaction with E222. This may favor the conformation of TKT, where E222 interacts with K16, as in 3OOY.

As shown in Figure 2C, K102^A^ (subunit A) is also distant from the active site: The closest active site residue R474^B^ (subunit B), which is involved in substrate binding, is 13.4 Å away from K102^A^. However, K102^A^ is at the entrance to the active site. Acetylation of K102 in TKT may therefore affect electrostatic interactions of the TKT substrates/products entering/exiting the active site. While in both TKT structures (3OOY and 4KXX), the K102^A^ residue is within the H-bond distance to the oxygen atom of A32^A^ of the same subunit, and it is also 5.5 Å from the E600^B^ residue of the other subunit. Thus, acetylation of K102^A^ may also affect the subunit interactions.

The TKT residue K499 is located in a small loop comprising four charged residues—K497, K499, D501, and R530 (Figure 2D). While Nε of the R530 guanidino group forms an electrostatic interaction with D501, the amino group of deacylated K499 is positioned appropriately for sharing the proton with NH_2_ of the R530 guanidino group. The K499 malonylation within this loop would create a tetrade of residues forming two salt bridges instead of the positively charged cluster on the TKT surface.

### 2.3. Correlation Analysis of the Interplay between the Levels of TKT Expression, Activity, Acylations, and Sirtuin 5

As shown earlier, adaptations to metabolic perturbations involve changes in the metabolic network, which are manifested in correlations between the parameters involved with addressing the perturbations [22,24,25]. In our study, we assess the role of the varied levels of TKT expression, activity, acylations, and sirtuin 5 in the metabolic perturbations induced by OT or M + A. The correlations between these parameters responding to the experimental perturbations in the thiamine-dependent processes are compared to each other (Table 1A) and to the correlations inherent in the control state (Table 1B).

Several relationships between the studied parameters are inherent in both the control and perturbed states, manifesting their basic significance. First of all, this is demonstrated by the expectedly strong positive correlations between the endogenous holoenzyme of TKT (assessed by the TKT activity without added ThDP) and the total level of active TKT (assessed by the TKT activity with added ThDP), observed in all the states (Table 1). Furthermore, in all the states, the acetylation of K6 or expression of TKT positively correlates with the expression of sirtuin 5. Moreover, the TKT–sirtuin 5 correlation becomes more significant in the perturbed states (Table 1A), presumably an indicator of the adaptive role of the relationship.

Paradoxically, TKT protein expression does not correlate positively with the TKT activity in either the control or perturbed states. Moreover, in the M + A model, the TKT expression negatively correlates with the TKT activity (Table 1). These findings stress the role of posttranslational regulation of TKT activity.

Indeed, the correlations of the TKT acetylation undergo significant and perturbation-specific changes, compared to the control state. A strong negative correlation between the acetylation level of K102 and sirtuin 5 protein and a low negative correlation between the acetylations of K102 and K6 are observed in the control state (Table 1B) but disappear in the two perturbed states (Table 1A). In the M + A experiment, the loss of the negative relationships of the acetylated TKT K102 with sirtuin 5 or acetylated K6, inherent in the control animals (Table 1B), is accompanied by the appearance of the negative correlations of the acetylation levels of K102 and, to a lesser extent, of K6, with TKT activity (Table 1A). In the OT series, significant negative correlations of the TKT activity with the levels of K6 acetylation and K499 malonylation become evident (Table 1A) under the lost relationships between the TKT acetylations and sirtuin 5 (Table 1B). As a result, the metabolic perturbations change the interplay between the TKT acylations and sirtuin 5, which is associated with a negative impact of the acylations on the TKT activity.

Altogether, the correlation analysis reveals a basic relationship between the protein expression of TKT and sirtuin 5, which is involved in the control of the TKT activity by acylation of its lysine residues. The metabolic perturbations, especially that by OT, increase the correlation between the expression of TKT and sirtuin 5 proteins, pointing to a heightened relationship between the two proteins under the perturbation of the thiamine-dependent metabolism. As a result, the perturbations induce a disbalance in the acylation system, supported by the negative impact of the TKT acylations on the enzyme activity (Table 1). Although the different acylations of TKT are highly interdependent, the inhibitors of the thiamine intracellular transport (M + A) or the ThDP-dependent enzymatic functions (OT) cause the perturbation-specific shifts in the regulation of TKT function by site-specific acetylation and malonylation.

## 3. Discussion

This work reveals for the first time that metabolic perturbations upon administration of the inhibitors of intracellular thiamine transporters (M + A) or ThDP-dependent enzymes (OT) to rats are manifested in the posttranslational acylations of the brain TKT. The regulation of TKT by acylation of its lysine residues agrees with our observation of a correlated expression of TKT and sirtuin 5, which is heightened in the metabolically challenged animals. Moreover, different types, i.e., malonylation and acetylation, and sites, i.e., K6, K102, and K499, of the brain TKT acylation specifically respond to the inhibitors of the different thiamine-dependent processes (M + A or OT). Only when the function of the ThDP-dependent enzymes including the acyl-CoA producing 2-oxo acid dehydrogenases is strongly perturbed, i.e., upon OT administration, an increase in TKT malonylation is observed, accompanied by decreased expression of the demalonylase sirtuin 5 (Figure 1). The TKT acetylation undergoes subtle changes revealed by correlation analysis, in both models. The changes induce negative correlations between the TKT activity and the TKT acetylation at either K6 (the action of OT) or K102 (the action of M + A) residues.

### 3.1. The Brain TKT Regulation by Malonylation

Malonylation of lysine residues in proteins faces certain challenges for its identification due to neutral loss during fragmentation [26]. The ensuing lower stability of malonylation, compared to acetylation, results in rather weak signals of malonylated positions in the MS/MS spectra (Appendix A [2]). In our work, CID (collision-induced dissociation) fragmentation is used instead of the HCD (higher energy collisional dissociation) fragmentation, which better prevents neutral losses of fragile side chain bonds. This feature may have favored our identification of the previously unknown malonylation of the TKT residue K499. Yet in human and mouse samples, this residue is known to be acylated in the acetylation and ubiquitination reactions [27,28,29,30,31,32].

Our correlation analysis indicates that the level of the brain TKT K499 malonylation in the OT-treated animals increases along with decreasing activity of TKT (Table 1A). In good agreement with this result, cellular incubation with malonyl-CoA inactivates endogenous TKT inside the colorectal cancer cells [1]. K499 belongs to the TKT central domain, which participates in the active site formation [5,33]. Given the distant position of K499 from the active site (Figure 2), the negative correlation between K499 malonylation and TKT activity (Table 1A) suggests no causal relationship between the two correlated parameters. Although an indirect involvement of K499 in the TKT regulation cannot be excluded, K499 malonylation may be just a marker of other events associated with the lower TKT activity. For instance, malonylation of TKT may also occur at other lysine residues, within or near the active site, but these peptides are not detectable in our assay. Using the sirtuin 5 silencing and overexpression of TKT mutants [1], malonylation of TKT K281 has been suggested to cause a decrease in the TKT activity. Unfortunately, the relevant TKT malonylation intensities in this study lack quantification and are presented without statistical analysis, which interferes with a conclusive comparison of the TKT activity and K281 malonylation levels. Localization of K281 far away from the active site in the structures of human TKT (PDB 3OOY and 4KXX) does not support its functional role in the TKT substrate binding or catalysis. However, taking into account that both these malonylated sites in TKT, i.e., K281 and K499, occur within the patches having up to three positively charged residues, the TKT malonylation may well regulate the enzyme activity by controlling electrostatic interactions with other proteins or nucleic acids. Indeed, yeast TKT forms a stable complex with RNA [34,35]. TKT is also known to form a binary complex with glyceraldehyde-3-phosphate dehydrogenase [36] or the ternary complex between transketolase, transaldolase, and glyceraldehyde-3-phosphate dehydrogenase [37]. Such protein complexes may be physiologically important for glucose distribution between different pathways, involving the TKT activity adjustment according to metabolic demands. Interestingly, in the liver of mice with sirtuin 5 knockout, increased malonylation of glyceraldehyde-3-phosphate dehydrogenase decreases glycolytic flux [2]. Based on this finding, the authors suggest that in the liver, the glyceraldehyde-3-phosphate dehydrogenase malonylation controls the glucose flux distribution between glycolysis and PPP or glycogen synthesis. In contrast, in the colorectal cancer cells, the sirtuin 5 knockout and ensuing TKT malonylation inhibit the PPP-dependent production of ribose-5-phosphate in the non-oxidative branch of PPP [1]. Mutual regulation of the protein–protein interactions and malonylation in the complex between TKT and glyceraldehyde-3-phosphate dehydrogenase may thus be a key to the distribution of glucose between different pathways in a cell-specific manner. The interaction of mammalian TKT with glucose-regulated protein 78 (GRP78, also known as BIP) in colorectal cancer cells [38] provides further support for the role of the malonylation-regulated, protein–protein interactions of TKT in the control of glucose fluxes.

### 3.2. The Brain TKT Regulation by Acetylation

Our study shows that the TKT K6 and K102 residues undergo acetylation in the brain. To the best of our knowledge, this is the first time when the acetylated TKT residues are identified in the mammalian brain. Earlier identifications of these acetylated residues mostly refer to cell lines [39,40,41]. TKT K6 acetylation is also observed in liver tissue [27,28]. Given the known dependence of the protein acylation patterns on cells and tissues, identification of the protein acylation sites in different tissues is important to understand molecular mechanisms of the tissue-specific regulation [42,43].

The K6 and K102, acetylated in the brain TKT, belong to the enzyme N-terminal domain participating in the active site formation. Although the residues are far away from the active site (Figure 2), their acetylation may change electrostatic interactions within the N-terminal domain and close to the subunits interface. The acetylation-regulated formation of the alternative salt bridges between E222 and either K6 or K16 may affect the conformational state of the TKT N-terminus. In turn, the conformational shifts within the active-site-forming N-terminal domain may affect the TKT activity by further changing the conformation of the active site. Furthermore, structurally disordered protein parts, such as the disordered N-terminus of TKT, often participate in protein–protein interactions, potentially affecting enzyme activity. A less exposed to the solvent, K102 residue at the entrance to the active site may be involved in the acetylation-dependent control of the substrate access to the active site or the product exit. The residue acetylation may also affect interactions between the TKT subunits, which are significant for enzyme catalysis [44,45]. All these factors may underlie the negative correlation between the K102 acetylation and TKT activity (Table 1A, M + A).

### 3.3. Biological Implications of TKT Acylation

Recent data on regulation of the TKT proteolytic degradation suggest the role of acetylation in enzyme stabilization. In HeLa and RPMI8226 cells, acetylation of TKT K6 residue results in a slower protein degradation, compared to the non-acetylated TKT [39]. The TKT residues K6, K102, and K499 are also known to be ubiquitinated [29,30,31,32,39]. In view of the ubiquitination-directed proteolysis, the competitive acetylation of the same TKT residues may prevent such proteolysis. However, the negative correlations of the TKT activity and acetylation do not testify to the decreased proteolysis of the active (in standard assay) TKT. On the other hand, the existence of the TKT pool, which is not catalytically active in the assayed TKT reaction, is supported by no correlation between the TKT expression and activity, even when the total activity is determined in the presence of ThDP (Table 1). Moreover, administration to animals of the inhibitors of thiamine transporters induces a negative correlation between the TKT expression and the assayed TKT activity (Table 1A). Altogether, the data suggest that the TKT acetylation may be linked to some “moonlighting” [46] function of TKT, inactive in its traditional catalytic reaction. Indeed, TKT from yeast is inactive in its complex with RNA, although the biological significance of this complex is unknown [35]. Transketolase of *Escherichia coli* is a transcription factor [47]. The TKT binding to nucleic acids, usually employing electrostatic interactions, is especially interesting in regard to the protein acetylation. Other moonlighting functions of TKT are linked to enzyme secretion. Secretion of TKT by *Entamoeba histolytica* discriminates between the pathogenic and non-pathogenic strains [48]. TKT is also known to be localized on the cellular surface of *Mycoplasma pneumoniae*, where its interaction with the host plasminogen determines the success of the long-term colonization of the human respiratory system [49]. In mammals, TKT is known to translocate to the nucleus [10,50]. In ischemic heart failure, nuclear TKT binds to poly(ADP-ribose) polymerase 1, facilitating the polymerase cleavage, and activates apoptosis-inducible factor [50]. One cannot therefore exclude that TKT inactive in its traditional reaction may be expressed for some moonlighting functions in the brain. The contribution of this inactive TKT to the total TKT protein, quantified by our MS analysis, is suggested by the absence or even a negative correlation between the brain TKT expression and activity in the transketolase reaction (Table 1).

## 4. Materials and Methods

### 4.1. Reagents

When not specified otherwise, chemicals were obtained from Merck (Dia-m, Moscow, Russia). Metformin (Merck, #D150959, Dia-m, Moscow, Russia), amprolium (Merck, #A0542, Dia-m, Moscow, Russia), and oxythiamine (Merck, #O4000, Dia-m, Moscow, Russia) were dissolved at the day of injections, and pH was adjusted to neutral value with NaOH, if necessary. The barium salt of phosphopentose mixture containing xylulose 5-phosphate and ribose 5-phosphate was prepared using an earlier described method [51]. The barium salt was converted to potassium salt before use with Dowex 50W-X8 in H+-form (Merck, #1.05221, Dia-m, Moscow, Russia). α-Glycerophosphate dehydrogenase-triosephosphate isomerase mixture was obtained from Merck (#G1881, Dia-M, Moscow, Russia). Deionized MQ-grade water was used for solution preparations.

### 4.2. Animal Experiments

All animal experiments were performed according to the Guide for the Care and Use of Laboratory Animals published by the European Union Directives 86/609/EEC and 2010/63/EU and were approved by the Bioethics Committee of Lomonosov Moscow State University (protocol 139-a-2 from 19 May 2022).

Fifty-one Wistar (RRID: RGD_13508588) male rats (approx. 5–6 weeks and 158.7 ± 11.0 g at the beginning of the experiment, 9–10 weeks and 304.7 ± 25.9 g at the end of the experiment) were used in the two independent animal series. Each series contained an experimental group and a control group. Oxythiamine was administered to rats (n = 16, one died) as a single i.p. injection with a corresponding control group (n = 11) receiving equal volumes of physiological solution (0.9% sodium chloride). A combination of metformin and amprolium was administered to rats (n = 12, one died) as two separate injections. The control group (n = 12) received the two injections of physiological solution. In the MS experiment, up to seven samples were lost due to technical reasons in the series with metformin and amprolium; therefore, the number of animals for each of the MS-detected parameters was decreased as indicated in the figures.

The injections were made in the morning (ZT 2 ± 1) for 30 days long without breaks. Oxythiamine was administered at a dose of 1.5 mg per kg. Metformin was administered at a dose of 150 mg per kg and amprolium at a dose of 40 mg per kg. Doses were selected based on the published data and preliminary experiments to achieve significant effects with minimal lethality [52,53,54,55]. Doses between species were recalculated using a formula recommended by FDA [19]. Two rats—one from the group with the combination of metformin and amprolium and one from the oxythiamine group—died before the end of the study on days 9th and 16th, respectively. Twenty-four hours after the last injections, the animals were killed by decapitation. The brains were excised and transferred on ice, where the cortices were separated to be frozen in liquid nitrogen 60–90 s after decapitation. The cortices were stored at −70 °C.

### 4.3. Homogenization of Rat Brain Tissue

The brain cortex tissue (0.5 g) was homogenized for 2 min in 1.25 mL of the homogenization buffer (50 mM MOPS, pH = 7.0, containing 2.7 mM EDTA, 20% glycerol and protease inhibitors: 0.2 mM AEBSF, 0.16 µM aprotinin, 3.33 µM bestatin, 3 µM E-64, 2 µM leupeptin, and 1.4 µM pepstatin A), using an ULTRA-TURRAX^®^ T-10 Basic disperser (IKA, Staufen, Germany) at the speed parameter “3” as described before [24]. The homogenates were sonicated by Bioruptor^®^ (Diagenode, Liege, Belgium) for 7 cycles of 30 s sonication and 30 s pause at the sonicator low intensity.

### 4.4. TKT Activity Assay

The catalytic activity of TKT in the cerebral cortex homogenates was measured spectrophotometrically using CLARIOstar Plus microplate reader (BMG Labtech, Ortenberg, Germany) in the reaction:
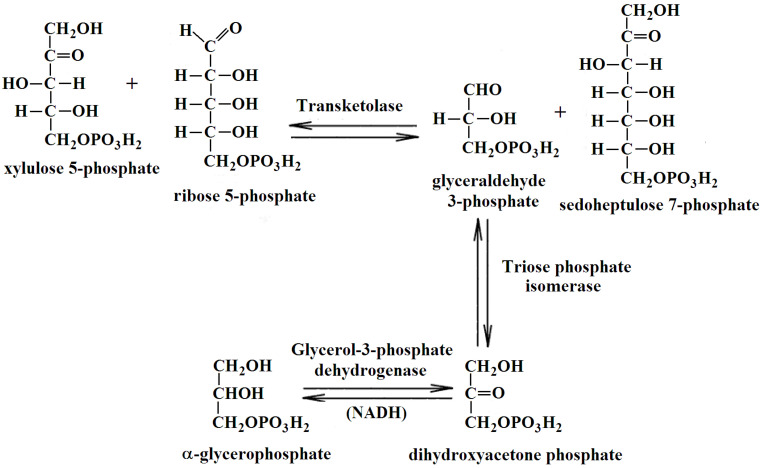


The rate of NADH oxidation in the coupled system shown above was measured using an earlier described method [56]. TKT was measured in an aliquot of each homogenate, preincubated with an equal volume of the 50 mM glycylglycine buffer with 3 mM CaCl_2_, pH 7.6, on ice for at least 30 min before measurement. The reaction mixture contained 50 mM glycylglycine, 2.5 mM MgCl_2_, 0.25 mM NADH, 3 U/mL of triosephosphate isomerase and glyceraldehyde phosphate dehydrogenase, 3 mg/mL potassium salt of the phosphopentose mixture, pH 7.6, and 0.2 mM ThDP (to assay total level of active TKT in the homogenates) or no ThDP (to assay endogenous level of holoTKT in the homogenates). The samples were preincubated in the reaction mixture without phosphopentose solution for 30 min. The reaction was initiated by the addition of phosphopentoses. The blank was assayed omitting phosphopentoses.

### 4.5. Procedure of LC-MS/MS

The polyacrylamide gels were subjected to in-gel digestion and Nano-LC-MS/MS analysis as described earlier [19]. The spectra acquisition consisted of an orbitrap full MS scan (FTMS; resolution 60,000; m/z range 400–2000) followed by up to 15 LTQ MS/MS experiments (Linear Trap; minimum signal threshold: 500; dynamic exclusion time setting: 30 s; singly charged ions were excluded from selection, normalized collision energy: 35%; activation time: 10 ms). Raw data processing, protein identification, and phosphorylation assignment of the high-resolution orbitrap data were performed by PEAKS Studio Xpro (Bioinformatics Solutions, Waterloo, ON, USA). The false discovery rate was set to <1%.

### 4.6. Quantification of the TKT Acylations, Expression, and Sirtuin 5 Protein Level

Relative MS-based quantification of the TKT acylations, the expression of TKT, sirtuin 5 (Sirt5), and the proteins for the normalization procedure (beta-actin ActB and beta-tubulin Tubb3) was performed using the Skyline platform [57], as described previously [19,20]. Shortly, the peptide retention time provided in PEAKS Studio Xpro was used to identify the peptide in Skyline. The relative abundance of the reliably identified peptide was estimated by integrated peak area, determined for the peptide in Skyline. All the used peptides are listed in Appendix A (P1–P18) together with the corresponding modifications, masses, and charge parameters.

The relative protein abundances of TKT or sirtuin 5 in each sample were calculated using the sum of the integrated peak areas of the unique TKT (P6–P9) or sirtuin 5 (P16–P18) peptides, as shown in Appendix A. For each sample corresponding to a single animal, the TKT or sirtuin 5 abundances were normalized to the sum of integrated peak areas of beta-actin and beta-tubulin (ActB and Tubb3, P10–P15, Appendix A). The normalized protein abundances were taken as estimates of the protein expression levels.

The relative levels of TKT K499 malonylation and K102 acetylation were determined as the ratio of the acylated and non-acylated peptide abundances, i.e., (P1 integrated peak area)/(P2 integrated peak area) and (P3 integrated peak area)/(P4 integrated peak area), correspondingly, in each sample. In the case of the K6-containing peptide, the non-acylated peptide was not identified. Hence, the sum of the integrated peak areas of the P6–P9 peptides of TKT was used for the normalization, i.e., the P5-integrated peak area was normalized to the sum of the integrated peak areas of the unique TKT peptides (P6–P9) in the same sample.

### 4.7. Structural Visualization

PyMOL v2.5 (PyMOL Molecular Graphics System, Schrödinger, LLC, New York, NY, USA) was used for the visualization of TKT. Variable conformational states of mammalian TKT were assessed employing the PDB-deposited TKT structures 3OOY (with ThDP) and 4KXX (with ThDP and hexahydroxyheptyl dihydrogen phosphate in the substrate site). Superposition of the structures was completed with the help of multiple structural alignments by PyMOL.

### 4.8. Statistics

Statistical analysis was performed using GraphPad Prism, version 8.0 (GraphPad Software Inc., La Jolla, CA, USA). Normal distribution was tested using D’Agostino and Pearson omnibus normality test (*p* < 0.05). Since not all the parameters were distributed normally, Spearman correlations were used for the correlation analysis. Differences between the two groups are analyzed using the Mann–Whitney U-test. The ROUT test for outliers [58] is applied using the default parameter of Q coefficient equal to 1%.

## 5. Conclusions

Posttranslational modifications of TKT by acylations are involved in the brain response to perturbed thiamine-dependent metabolism. Negative correlations of the TKT activity with acetylation of K6 and K102 residues and malonylation of K499 residue are observed. The residues are located on the enzyme surface distant from the active site. The acylations may affect the TKT function through changed electrostatic interactions, causing conformational shifts and/or regulating the TKT interactions with proteins or nucleic acids. Acetylation of K102 may modify the substrate access or product exit from the active site. Malonylation of K499 in TKT is significantly changed along with decreased sirtuin 5 expression upon the oxythiamine-perturbed function of ThDP-dependent enzymes. No positive correlation between the brain TKT expression quantified by MS, and the TKT catalytic activity in the ThDP-saturated transketolase reaction suggests a significant portion of the brain TKT to be inactive in the transketolase reaction.

## Figures and Tables

**Figure 1 ijms-25-00917-f001:**
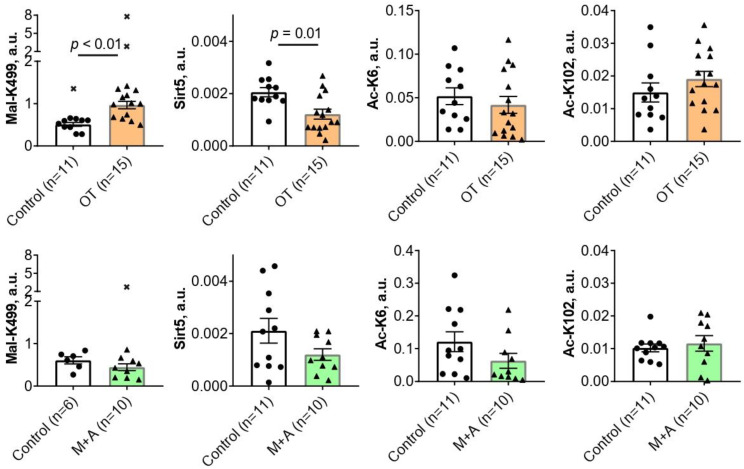
The effects of oxythiamine (OT, orange bars) or metformin and amprolium (M + A, green bars) on the levels of TKT acylations at the residues indicated on the ordinates, and sirtuin 5 protein (Sirt5) in the rat cerebral cortex. The studied animal groups with the animal number *n* are shown on the X axes. The analyzed parameters are indicated on the Y axes. Mal—malonylated; Ac—acetylated. Comparison of the experimental and control groups is achieved using the Mann–Whitney U-test. *p*-Values of the significant (*p* ≤ 0.05) differences are indicated on the graphs. All *p*-values are given in Appendix A, which also includes *p*-values according to the Student *t*-test for comparison. Crosses denote excluded outliers determined by the ROUT test as described in “Materials and Methods”. Each data point corresponds to an animal. The number of animals in the groups is indicated in the X-axis legends. a.u.—arbitrary units.

**Figure 2 ijms-25-00917-f002:**
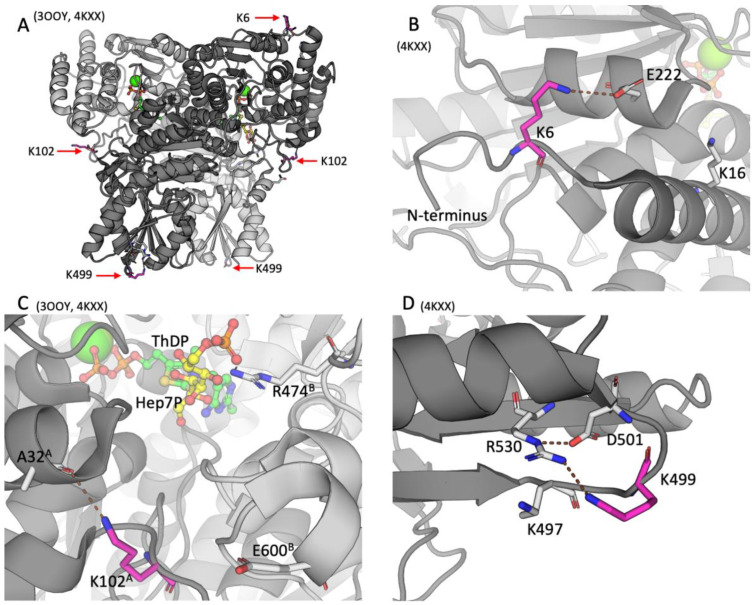
The acylation sites of TKT in the rat cerebral cortex. IDs of the used PDB structures are indicated above each of the four panels. TKT is represented as a cartoon model with different subunits shown in the light and dark shades of gray. The studied lysine residues undergoing acylations are shown in magenta; the substrate analog hexahydroxyheptyl dihydrogen phosphate (Hep7P) and ThDP are shown in yellow and green, respectively. Other residues or ligands are shown in gray. Standard color code is used for ions and non-carbon atoms. Brown dashed lines show the H-bonds discussed in the text. (**A**) Overall structure of TKT dimer. (**B**) The K6 residue of the N-terminal domain of TKT. (**C**) The K102 residue of the N-terminal domain of TKT at the entrance to the active site. The superscripts of the indicated residues denote their belonging to one of the subunits (**A** or **B**) in the TKT dimer. (**D**) The K499 residue in the central domain of TKT.

**Table 1 ijms-25-00917-t001:** Correlations between the levels of TKT expression, sirtuin 5 expression, TKT acylations, and TKT activity. Every cell contains a Spearman correlation coefficient (above) and its *p*-value (below). The cells presenting the correlations with significant (*p* < 0.05) *p*-values are in bold and colored in red for the positive correlation coefficients or blue for the negative ones. The cells presenting the correlation trends (0.05 < *p* < 0.1) are in italics and a lower color intensity. (**A**) The correlations of the indicated parameters determined for the pooled animals studied in the experiments with metformin and amprolium (M + A, top right, n = 16–23) or oxythiamine (OT, bottom left, n = 21–26). (**B**) The correlations of parameters determined in the control animals (n = 15–22). Mal—malonylated; Ac—acetylated; expr—protein expression; Act—activity.

**A. Perturbations in Thiamine Metabolism**								
	M+A	Mal-K499	Ac-K6	Ac-K102	TKT-expr	Act -ThDP	Act +ThDP	Sirt5
OT								
Mal-K499			−0.14	−0.15	0.20	−0.07	−0.22	−0.19
			0.66	0.62	0.51	0.80	0.43	0.53
Ac-K6		−0.22		−0.08	0.22	*−0.35*	*−0.38*	**0.76**
		0.31		0.74	0.35	*0.12*	*0.09*	**0.00**
Ac-K102		0.03	0.13		0.26	**−0.45**	*−0.41*	−0.11
		0.88	0.54		0.25	**0.04**	*0.07*	0.65
TKT-expr		−0.08	−0.04	0.12		**−0.45**	**−0.47**	**0.59**
		0.73	0.84	0.55		**0.04**	**0.03**	**0.01**
Act -ThDP		**−0.52**	**−0.47**	−0.10	0.15		**0.95**	−0.26
		**0.01**	**0.02**	0.62	0.47		**0.00**	0.26
Act +ThDP		*−0.40*	**−0.50**	−0.20	−0.08	**0.97**		−0.30
		*0.07*	**0.01**	0.35	0.72	**0.00**		0.19
Sirt5		−0.16	**0.60**	−0.18	**0.51**	−0.21	−0.30	
		0.47	**0.00**	0.38	**0.01**	0.31	0.15	
**B. Control State**								
		Mal-K499	Ac-K6	Ac-K102	TKT-expr	Act -ThDP	Act +ThDP	Sirt5
Mal-K499			−0.12	−0.26	0.32	−0.31	−0.36	0.18
			0.68	0.34	0.24	0.23	0.19	0.52
Ac-K6				*−0.36*	−0.20	−0.16	*−0.25*	**0.61**
				*0.10*	0.37	0.49	*0.28*	**0.00**
Ac-K102					0.04	0.07	*0.02*	**−0.52**
					0.88	0.76	*0.94*	**0.01**
TKT-expr						0.07	−0.10	*0.40*
						0.76	0.67	*0.06*
Act -ThDP							**0.93**	−0.10
							**0.00**	0.67
Act +ThDP								−0.14
								0.53
Sirt5								


## Data Availability

The data presented in this study are available in this article and Appendix A.

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
