# Peer review of "Posttranslational Acylations of the Rat Brain Transketolase Discriminate the Enzyme Responses to Inhibitors of ThDP-Dependent Enzymes or Thiamine Transport"

_ijms, 2024, doi:10.3390/ijms25020917_

Round 1

Reviewer 1 Report

Comments and Suggestions for Authors

The manuscript by Victoria I. Bunik and colleagues describes new findings that suggest a key role played by post-translational modifications of rat brain transketolase and the enzyme's activity. Specifically, using a proteomic approach to analyze the brains of rats subjected to oxythiamine treatment and control animals, the authors identified two acetyl-lysines (K6 and K102) and one malonyl-lysine (K499). The abundances of these modified amino acids in treated and control rats were compared; the effect of these modifications on transketolase activity was also studied.

The manuscript is based on a limited number of data and experiments and it would certainly be necessary to carry out further validations for the work to be suitable for publication.

In particular, as far as I know, it is the first time that the malonylation of lysine 499 has been described. However, the data provided by the authors do not allow us to understand how reliable this identification is. It would be appropriate to show the mass spectra that led to this identification. Furthermore, the peptide containing this modified amino acid also has a second lysine in its sequence that is not hydrolyzed by trypsin. How did the authors confirm that the modification involved K499? In fact, I know from experience that during the fragmentation process the modifications to the lysines are often lost, making it difficult to define the actually modified AA.

The acetylation of K102 was also described for the first time in this manuscript, whereas several other modification reported in the literature were not observed. This point should be discussed in detail by the authors.

I am rather perplexed about the data reported in the first two panels of figure 1. It is in fact easy to see that, although the average value obtained for the control and the treated samples is evidently different, some values measured for the first and second group practically coincide. A Student's T test analysis of these data would certainly give a much higher p value than that reported by the authors. Besides, since the accurate quantification of peptides by MS is not as obvious as that of small molecules, the method used to measure the values reported in the figure must be described in detail in the experimental section.

Author Response

We thank the reviewer for the critical comments which we have fully addressed as described below, that greatly improved the manuscript.

«The manuscript is based on a limited number of data and experiments and it would certainly be necessary to carry out further validations for the work to be suitable for publication.»

We thank the reviewer for the advices regarding presentation of additional data, that we do as described below. However, let us disagree with the statement on the “limited number of data”. We describe the work on the two animal models with different chronic treatments, involving 51 animals. These animals have been analyzed regarding:  MS-based quantifications of the three posttranslational modifications, MS-based quantifications of the transketolase (TKT) and sirtuin expression, assays of TKT activity and TKT saturation with its coenzyme ThDP. Besides, we perform the structural and correlation analyses to assess functional consequences of the TKT modifications in vitro and in vivo. Thus, we combined a range of structural and functional approaches to characterize molecular features of the brain TKT and their significance in vivo. The results are presented in 8 graphs, two Tables and 4 structural pictures. The fact that we combine all these data within logically organized and ready for comparison modules, cannot be defined as “limited number of data”, based simply on the number of the figures and table. Finally, the manuscript is focused on the brain TKT, as the enzyme is very important for the glucose metabolism and its malignant transformation, while modern studies on the structure-function relationship in naturally expressed transketolase in tissue/cell-specific biological systems are indeed very limited.

In particular, as far as I know, it is the first time that the malonylation of lysine 499 has been described.

Overall, none of the modifications were identified in the brain TKT, and malonyl-K499 was not earlier determined in other systems either. In general, malonylation faces certain difficulties in its identification. We added more discussion on this modification in the section 3.1

It would be appropriate to show the mass spectra that led to this identification.

We added Supplementary Figure 1 to show the spectra with the accompanying fragment mass tables indicating the reliability of peptide identifications.

 Furthermore, the peptide containing this modified amino acid also has a second lysine in its sequence that is not hydrolyzed by trypsin. How did the authors confirm that the modification involved K499?

The fragment masses of the peptide indicated in the table below the MS/MS spectrum, unambiguously determine the site of the modification, even if the second site is present in the sequence. All b-ions (blue labels in spectrum and table) support the K499 position of the malonylation. Moreover, there are no hints for a modification of the second lysine, because all y-ions provide evidence for the unmodified second lysine.

 In fact, I know from experience that during the fragmentation process the modifications to the lysines are often lost, making it difficult to define the actually modified AA.

You are fully right, and probably it is the lower stability of malonylation, compared to acetylation, that results in rather weak signals in the MS/MS spectra due to neutral losses. We have used CID (collision-induced dissociation) fragmentation instead of the HCD (higher energy collisional dissociation) fragmentation. CID is softer and better prevents neutral losses of fragile sidechain bonds. Ouridentification was reliable and calculated by the PEAKS algorithm as the statistically significant one. Moreover, we can also account on the peptide precursor mass (MS1) to conclude on the modification and all described quantifications were made on this level. The corresponding discussion is added in the section 3.1.

The acetylation of K102 was also described for the first time in this manuscript, whereas several other modification reported in the literature were not observed. This point should be discussed in detail by the authors.

The K102 acetylation is known from databases, we added more on this. In particular, it is important to note that all the modifications of TKT which we identified, were sporadically reported in different systems, but not for the animal brain enzyme studied by us. That said, the tissue/cell specificity of the post-translational modifications is well known, stressing the need to pay attention to specific systems if one desires to establish the regulatory potential of the modifications in vivo. We already had a notion about other acylations in the previous version: “For instance, malonylation of TKT may also occur at other lysine residues, within or near the active site, but these peptides are not detectable in our assay.“

Overall, we would like to draw the attention of the reviewer to Table 1, where the quantified levels of the acylated peptides showed multiple significant correlations with independently determined TK activity and expression of TKT and sirtuin 5. These correlations strongly argue for biological significance of the determined acylation levels.

I am rather perplexed about the data reported in the first two panels of figure 1. It is in fact easy to see that, although the average value obtained for the control and the treated samples is evidently different, some values measured for the first and second group practically coincide. A Student's T test analysis of these data would certainly give a much higher p value than that reported by the authors.

The results of statistical analysis are often different from what is “easy to see”. Besides, Student’s test is known to often give better p values than Mann-Whitney test. To address this comment, we calculated the significances according to Student’s test, too. As seen from the added Supplementary Table S2, for the questioned cases this resulted in p-values for higher significances than obtained by Mann-Whitney test: for malonyl-K499 p=0.0002, for sirt5 p=0.003. However, in the main text, we leave the Mann-Whitney test, as this is usually recommended for the work involving animals, where the number of samples is limited by ethical considerations.

 Besides, since the accurate quantification of peptides by MS is not as obvious as that of small molecules, the method used to measure the values reported in the figure must be described in detail in the experimental section.

We have published several papers using this method, to which we refer here. We added more details to methods and more references.

Reviewer 2 Report

Comments and Suggestions for Authors

In this manuscript, Vasily A. Aleshin and colleagues studied posttranslational acylations of the rat brain transketolase that discriminate the enzyme responses to inhibitors of ThDP-dependent enzymes or thiamine transport. The overall manuscript is well structured and well written, and this study is novel. I have just a few concerns, as given below-

Line # 47: Please write the complete form here: "each of app. 70 kDa"

Please add references in the following sentences:

"TKT is a thiamine diphosphate (ThDP)-dependent enzyme of the non-oxidative branch of pentose phosphate pathway (PPP), where ribose-5-phosphate for cellular nucleotide biosynthesis is generated in the oxidative and non-oxidative branches."

"The electrostatics-affecting modifications of metabolic enzymes by acylations of their lysine residues, such as the long-known acetylation of histones, are suggested by recent discovery of an interplay between TKT and a NAD+-dependent deacylase of negatively charged acyls, sirtuin 5."

"Overall, the data on the cancer cells presented in this work indicate that malonylation of TKT inactivates the enzyme, that causes DNA damage due to insufficiency of ribose-5P."

"The accumulating data suggest that modification of TKT by the negatively charged acyls removed by sirtuin 5, may result in the observed TKT isoforms affecting the enzyme charge and function. The varied responses to the knockout observed in different cells, manifest dependence of the responses on specific homeostatic networks. However, the cell-specific regulation and metabolic impact of the posttranslational modifications of TKT have been characterized only in the artificially created sirtuin 5 knockouts."

The text should be checked for typos throughout. For example: "Oxythiamine (OT) is a classic inhibitor of these functions in vivo, as I living systems it is converted into oxy-ThDP, blocking the enzymatic active sites with high affinity [10]."

To make consistency, if possible, please add references at the end of a sentence. Example: "Another known thiamine antagonist, amprolium, is a coccidiostatic [14,15], which blocks intracellular transport of thiamine through ThTR-1 and OCT transporters [10]."

Figure 1: Please write the p values in every comparison even though they are not significant. Also, write the full form if any abbreviations are used. Example: a.u. 

Author Response

We are grateful to the reviewer for the positive estimation and critical comments which allowed us to improve the manuscript. Below, we provide our answers to specific comments.

Line # 47: Please write the complete form here: "each of app. 70 kDa"

 Please add references in the following sentences: 

 The text should be checked for typos throughout.

Done

To make consistency, if possible, please add references at the end of a sentence. Example: "Another known thiamine antagonist, amprolium, is a coccidiostatic [14,15], which blocks intracellular transport of thiamine through ThTR-1 and OCT transporters [10]."

 In this specific example and other similar ones, we provide the different references to the different statements, i.e. “amprolium is a coccidiostatic [14,15]” and “amprolium blocks intracellular transport of thiamine through ThTR-1 and OCT transporters [10]”. In this way, the reader may look for these distinct issues in the specified papers. In simple sentences with no ambiguity on the reference relation to the statement, we paid attention to the requested edits.

Figure 1: Please write the p values in every comparison even though they are not significant.

Excessive indication of p values on the graphs distracts the reader attention from the major findings. On the other hand, we agree with the reviewer that knowledge on the probability is important even if it does not fall into the usually accepted range for statistically significant events. Therefore, we provide Supplementary Table S2 where all the statistics are indicated. We did it according to the two tests, also to satisfy the comment of the other reviewer.

“…Also, write the full form if any abbreviations are used. Example: a.u.

 We deciphered a.u. in the figure legend, and paid attention to other abbreviations.